# First-Trimester Maternal Serum Adiponectin/Leptin Ratio in Pre-Eclampsia and Fetal Growth

**DOI:** 10.3390/life13010130

**Published:** 2023-01-03

**Authors:** Victoria E. de Knegt, Paula L. Hedley, Anna K. Eltvedt, Sophie Placing, Karen Wøjdemann, Anne-Cathrine Shalmi, Line Rode, Jørgen K. Kanters, Karin Sundberg, Ann Tabor, Ulrik Lausten-Thomsen, Michael Christiansen

**Affiliations:** 1Department for Congenital Disorders, Statens Serum Institut, 2300 Copenhagen, Denmark; 2Department of Paediatrics, University Hospital Slagelse, 4200 Slagelse, Denmark; 3Brazen Bio, Los Angeles, CA 90014, USA; 4Department of Gynecology and Obstetrics, Bornholm Hospital, 3700 Rønne, Denmark; 5Department of Obstetrics, Hillerød Hospital, 3400 Hillerød, Denmark; 6Department of Clinical Biochemistry, Copenhagen University Hospital Rigshospitalet, 2600 Glostrup, Denmark; 7Department of Biomedical Sciences, University of Copenhagen, 2200 Copenhagen, Denmark; 8Center of Fetal Medicine, Department of Obstetrics, Copenhagen University Hospital Rigshospitalet, 2100 Copenhagen, Denmark; 9Department of Clinical Medicine, Faculty of Medical and Health Sciences, University of Copenhagen, 2200 Copenhagen, Denmark; 10Department of Neonatology, University Hospital Rigshospitalet, 2100 Copenhagen, Denmark

**Keywords:** adipocytokine, adiponectin, A/L ratio, birth weight, leptin, metabolic syndrome

## Abstract

The serum adiponectin/leptin ratio (A/L ratio) is a surrogate marker of insulin sensitivity. Pre-eclampsia (PE) is associated with maternal metabolic syndrome and occasionally impaired fetal growth. We assessed whether the A/L ratio in first-trimester maternal serum was associated with PE and/or birth weight. Adiponectin and leptin were quantitated in first-trimester blood samples (gestational week 10^+3^–13^+6^) from 126 women who later developed PE with proteinuria (98 mild PE; 21 severe PE; 7 HELLP syndrome), and 297 controls, recruited from the Copenhagen First-Trimester Screening Study. The A/L ratio was reduced in PE pregnancies, median 0.17 (IQR: 0.12–0.27) compared with controls, median 0.32 (IQR: 0.19–0.62) (*p* < 0.001). A multiple logistic regression showed that PE was negatively associated with log A/L ratio independent of maternal BMI (odds ratio = 0.315, 95% CI = 0.191 to 0.519). Adiponectin (AUC = 0.632) and PAPP-A (AUC = 0.605) were negatively associated with PE, and leptin (AUC = 0.712) was positively associated with PE. However, the A/L ratio was a better predictor of PE (AUC = 0.737), albeit not clinically relevant as a single marker. No significant association was found between A/L ratio and clinical severity of pre-eclampsia or preterm birth. PE was associated with a significantly lower relative birth weight (*p* < 0.001). A significant negative correlation was found between relative birth weight and A/L ratio in controls (*β* = −0.165, *p* < 0.05) but not in PE pregnancies), independent of maternal BMI. After correction for maternal BMI, leptin was significantly associated with relative birth weight (*β* = 2.98, *p* < 0.05), while adiponectin was not significantly associated. Our findings suggest that an impairment of the A/L ratio (as seen in metabolic syndrome) in the first trimester is characteristic of PE, while aberrant fetal growth in PE is not dependent on insulin sensitivity, but rather on leptin-associated pathways.

## 1. Introduction

Pre-eclampsia (PE) is a serious pregnancy complication characterized by high blood pressure and proteinuria and/or impaired liver and kidney function, pulmonary edema, hematological complications, seizures, or uteroplacental dysfunction [1,2]. The prevalence of PE is 3–5% in the Western world and it is a major contributor to maternal and fetal/neonatal morbidity [1]. In places with limited access to healthcare, e.g., in large parts of Asia, Africa, and South America, between one-tenth and one-quarter of maternal deaths are associated with hypertensive disorders of pregnancy [3]. PE predisposes both mothers and offspring to a plethora of deleterious health outcomes. Mothers have an increased risk of developing hypertension and other cardiovascular diseases [4], metabolic syndrome, and overweight and obesity years after pregnancy [5]. Infants too are at risk of developing cardiovascular disease [6], metabolic syndrome and overweight and obesity later in life [6,7,8]. The pathophysiological background of PE is elusive, but most likely heterogeneous, involving uterine–placental dysangiogenesis, endothelial dysfunction, immunological abnormalities, genetic factors, or metabolic disturbances [9].

Pregnancy requires major metabolic adaptations to ensure adequate nutrient supply for the fetus [10]. An important aspect of this is changes in maternal insulin sensitivity (IS) and insulin resistance (IR) that is mediated by a range of maternal, placental, and fetal hormones [11]. The first trimester is characterized by a relative increase in insulin sensitivity, needed to ensure uptake of nutrients in the mother; this changes as insulin resistance develops during the second trimester [12,13].

Pre-pregnancy IR is associated with the development of gestational diabetes mellitus (GDM) [14], with an ensuing risk of fetal over-nutrition and macrosomia [15,16,17], as well as PE [18]. Early-onset PE is frequently associated with fetal growth restriction (FGR) [19], while late-onset PE is more often associated with macrosomia [2]. Both fetal under- and overnutrition may lead to changes in body composition in the fetus that persist after birth [20,21]. These, in turn, may lead to permanent metabolic alterations, epigenetic changes, and fetal programming, that ultimately result in the development of metabolic syndrome and its associated diseases later in life [6,7,8,22].

The risk of developing PE increases with increasing body mass index (BMI) [23]. Furthermore, adiponectin, exhibiting decreased maternal serum levels in pregnancy [24], and leptin, where the maternal serum levels increase through pregnancy to normalize after birth [25], play a major role in the metabolic adaptations to pregnancy. There is a small placental contribution to maternal leptin levels [26,27], but not to the extent that it obviates the relation between maternal BMI and leptin. A number of hormonal factors regulate the insulin signaling pathway and control the balance between IS and IR in order to accommodate pregnancy [11] as well as compensate for the metabolic consequences of overweight and obesity [28]. Several hormones have been suggested to be involved in first-trimester fetal growth, and placental growth hormone appears to be of particular significance [29]. In pregnant women with BMI values on the extreme ends of the scale, insufficient homeostatic compensatory mechanisms may result in presentation of disease symptoms. This process may be reflected in adipocytokine levels, already in first-trimester [28].

Adiponectin and leptin are involved in a wide range of physiological processes and seem perturbed in GDM and PE [14,30,31,32,33,34,35,36,37]. Serum adiponectin/leptin ratio (A/L ratio) is a surrogate marker of IS that correlates well with signs of metabolic syndrome in childhood and adolescence [38,39]. It has been shown to correlate with insulin resistance in pregnancy [40]. However, it has not been associated with adverse pregnancy outcomes. 

In this study, we used A/L ratio as a marker of IS and hypothesize that the A/L ratio is perturbed in the first trimester of pregnancies that later develop PE. Furthermore, we assessed the significance of the A/L ratio for fetal growth, because maternal, placental, and fetal leptin have been reported to contribute to intrauterine growth [26]. Accordingly, this study examined the relationship between first-trimester A/L ratio and PE, as well as the relationship between adiponectin, leptin, and A/L ratio and birth weight. 

## 2. Materials and Methods

### 2.1. Study Design and Participants

This was a case–control sub-study of the 1997 to 2001 Copenhagen First-Trimester Screening Study focusing on screening for chromosomal disorders [41]. Several sub-studies on fetal growth [29], PE [27,42,43,44], and sources of Down syndrome serum marker variability [45,46], have been performed. The adipocytokine data used were collected from 2003 to 2010 and stored in an anonymized database. Only singleton pregnancies with first-trimester blood samples (gestational age (GA) 10^3^–13^6^) were included. GA was determined by crown rump length (CRL). Selection of cases and controls is described in Laigaard et al. [42]. Briefly, 126 pregnancies that developed PE were selected for the study and 297 controls were matched with respect to maternal age, parity, and GA at time of sampling [42]. Of the 126 women with PE, 98 had mild PE, 21 had severe PE, and seven had HELLP syndrome. Ten pregnancies ended in delivery before GA 34^+0^. Regarding ethnicity, 93% of the women were of North European descent. Demographic and clinical parameters are given in Table 1.

### 2.2. Data Collection

Detailed clinical information, as well as pregnancy-associated plasma protein A (PAPP-A) and beta-chorionic gonadotropin (hCGβ) concentrations, were collected as part of the Copenhagen First-Trimester Screening Study, as previously described [41]. PE pregnancies were categorized into one of three severity groups: mild PE, severe PE, and HELLP syndrome as per criteria from The International Society for the Study of Hypertension in Pregnancy (ISSHP) during the course of the First-Trimester Screening Study [47]. Briefly, PE was defined as persistent hypertension (either a systolic blood pressure ≥ 140 mm Hg or a diastolic blood pressure ≥ 90 mm Hg), occurring after 20 weeks of gestation in a previously normotensive woman, in combination with proteinuria (≥0.3 g in a 24 h urine collection or dipstick urine analysis of ≥1+). Severe PE was defined by a diastolic blood pressure > 110 mm Hg in combination with subjective symptoms and/or abnormal laboratory findings. HELLP syndrome was defined by the presence of hemolysis, elevated liver enzymes, and low platelets. Early PE was defined as a PE pregnancy that resulted in delivery prior to GA 34^+0^.

### 2.3. Biochemical Measurements

All blood samples were collected in dry containers and kept at 4 °C for a maximum of 48 h until storage at −20 °C. Adiponectin and leptin concentrations were measured in singlo following appropriate sample dilution [27,28] using the Human Adiponectin Enzyme-Linked Immunosorbent Assay (ELISA) development kit, Duo Set (DY1065), and the Human Leptin ELISA development kit, Duo Set (DY398), R&D Systems, Minneapolis, USA. The functional detection limits of the assays were 62.5 pg/mL (for adiponectin) and 31.25 pg/mL (for leptin).

Evaluation of pre-analytic variables demonstrated that both adiponectin and leptin were stable for at least 48 h at 23 °C and 10 freeze–thaw cycles. The intra-assay and inter-assay coefficients of variation were <5% and <10% for adiponectin and leptin, respectively.

### 2.4. Data Analysis

Birth weights corrected for GA, relative birth weight, were calculated using intrauterine growth curves developed by Marsal et al. [48]. Weight classifications of small-for-gestational-age (SGA), appropriate-for-gestational-age (AGA), and large-for-gestational-age (LGA) were calculated using ±24% as the upper and lower limits of relative birth weight [48]. The A/L ratio was calculated as ([adiponectin]/[leptin])/1000. Data were analyzed using generalized linear models following logarithmic transformation as appropriate. When necessary, non-parametric statistics were employed. A PAPP-A value corrected for GA at time of sampling was calculated using multiples of the median (MoM). MoM of logPAPP-A in both controls and PE pregnancies were calculated by performing a regression analysis in the control pregnancies of log_10_ PAPP-A on GA at time of sampling. All analyses were conducted using IBM SPSS Statistics, Version 28.0; Armonk, NY, USA: IBM Corp.

### 2.5. Ethics

The Copenhagen First-Trimester Study was approved by the Scientific Ethics Committee for Copenhagen and Frederiksberg Counties (No. (KF) 01-288/97) and the Data Protection Agency approved the study protocol. All participants gave written informed consent. 

## 3. Results

### 3.1. Comparison of PE and Control Pregnancies

The A/L ratio was lower in PE pregnancies, 0.17 (IQR: 0.12–0.27) than in controls, 0.32 (IQR: 0.19–0.62) (*p* < 0.001) (Table 1). PAPP-A was significantly lower in PE pregnancies than controls (*p* < 0.001) (Table 1). Furthermore, a significant association was found between A/L ratio and PAPP-A (log-transformed and controlled for GA at time of testing) in PE (β = 1.432, 95% CI = 0.057 to 2.807). hCGβ concentrations were not significantly different between the two groups (Table 1). Offspring of women with PE had a lower GA at birth and lower relative birth weight compared with the control group (*p* < 0.001) (Table 1). A larger proportion of PE pregnancies delivered SGA infants, 14.3% in PE vs. 2.0% in controls (*p* < 0.001) (Table 1). Compared with control pregnancies, women with PE had a significantly higher BMI (*p* < 0.001) (Table 1). CRL and nuchal translucency measurements did not differ significantly between PE pregnancies and controls (Table 1). Of the PE deliveries, 74% were induced compared with 16% in controls (*p* < 0.001). 

Multiple logistic regression showed that PE was negatively associated with log A/L ratio independent of maternal BMI (OR = 0.315, 95% CI = 0.032 to 0.214). Adiponectin (area under the curve (AUC) = 0.632) and PAPP-A (AUC = 0.605) were negatively, and leptin (AUC = 0.712) was positively associated with PE. However, the A/L ratio was a better predictor of PE (AUC = 0.737) (Figure 1). 

### 3.2. Comparison of Mild PE, Severe PE, and HELLP Syndrome

The demographic, clinical, and paraclinical characteristics of mild PE, severe PE, and HELLP pregnancies are presented in Table 2. There was no discernable difference between A/L ratios in the different severity groups (Figure 2). The proportion of early PE (<GA week 34) increased with increasing severity of PE (*p* < 0.001). There was no significant difference in relative birth weight between severity groups. However, the proportion of SGA infants was greater in severe PE and HELLP (*p* = 0.006).

### 3.3. Comparison of Early and Late PE 

Demographic, clinical, and paraclinical characteristics of early and late PE are presented in Table 3. No significant difference in A/L ratio between early and late PE was found. More frequently, early PE pregnancies were classified as severe PE and HELLP syndrome compared with late PE pregnancies, 80.0% and 17.2%, respectively (*p* < 0.001). Early PE infants had a significantly lower relative birth weight than late PE infants (*p* < 0.001). There was a difference in size for GA between early and late PE, where 80.0% were SGA in early PE and 8.6% were SGA in late PE (*p* < 0.001). 

### 3.4. Relative Birth Weight and A/L Ratio

A significant negative correlation was found between relative birth weight and A/L ratio in controls, but not in PE pregnancies (*β* = −0.165, *p* < 0.05). This was independent of maternal BMI. The relationship was also valid when only considering AGA controls, again independent of maternal BMI. After correction for maternal BMI, leptin was significantly associated with relative birth weight (*β* = 2.98, 95% CI = 2.361 to 14.353), while adiponectin was not significantly associated. No association was found between A/L ratio and CRL in either PE or control pregnancies, independent of maternal BMI.

## 4. Discussion

In this study, we showed, for the first time, that the A/L ratio in the first trimester was significantly lower in pregnancies that developed PE compared with controls. Previous studies have either not examined first-trimester pregnancies [40,49,50,51], or examined the effect of the A/L ratio in first-trimester obese women who developed PE [28]. In the latter study, a decrease in A/L ratio, albeit insignificant, was found [28]. The A/L ratio was, however, associated with the development of GDM in the same cohort [51]. This is compatible with the inverse correlation between the A/L ratio and the homeostasis model assessment of IR (HOMA-IR) in pregnancy [40]. The A/L ratio decreased with maternal BMI, a known risk factor of PE [28], but the A/L ratio outperformed maternal BMI as a predictor of PE. This is compatible with the A/L ratio being a diagnostic marker of PE in late pregnancy [52]. 

The leptin/adiponectin ratio (a marker of insulin resistance) is associated with cardiometabolic risk factors in children [38,39]. Furthermore, in adults, a decreased A/L ratio has been shown to be associated with IR in various metabolic disorders, including diabetes mellitus [53,54], polycystic ovary syndrome (PCOS) [55,56,57], metabolic syndrome [58], atherosclerosis [59], and obesity [60]. The reduction in A/L ratio thus likely reflects an increasing IR. This could also explain the association between PE and PCOS [61]. 

In first-trimester PE pregnancies, the change in A/L ratio is driven by both a decrease in adiponectin and an increase in leptin. Adiponectin is a protective factor against cardiometabolic diseases [62]; thus, the association between PE and later hypertension and diabetes type 2 could be explained by this relation. Adiponectin is not synthesized by the placenta; therefore, the changes in adiponectin concentrations are a result of physiological changes in the mother. Leptin is partially synthesized by the placenta and largely by adipose tissue in the mother [35]. However, the effect of leptin is modulated by the presence of soluble leptin receptor as well as the variable expression of leptin receptors in different tissues [63]. Pregnancy requires metabolic adaptations through gestation, mediated by several endocrine axes; therefore, the present study precludes a more precise analysis of the causation of changes in adiponectin and leptin. 

The receiver operating characteristic (ROC) curve analysis showed that the A/L ratio was a better predictor of PE than leptin and adiponectin individually. However, the AUC of the A/L ratio is so small that it precludes using the A/L ratio as a first-trimester single marker of later PE. Likewise, PAPP-A is a marker for PE, but only used as a single marker in high risk pregnancies. However, it is possible that the A/L ratio might contribute to improving the efficiency and clinical utility of either PE screening algorithms based on clinical information [64] or algorithms including ultrasound and biochemical markers [65]. An advantage of a biochemical marker, which may be measured on dried filter paper blood spots, is that it can be used in populations with challenged access to prenatal care. This might be of particular importance in low- and middle-income countries. 

There were no discernable differences between A/L ratios in the different severity groups of PE, or between early and late PE. Thus, the A/L ratio in the first trimester cannot be used to discern the severity or early delivery in PE pregnancies. This suggests that IR predisposes to PE, but other factors determine clinical presentation. A significantly larger cohort of PE cases will be needed to further investigate the performance of the A/L ratio in early- and late-onset PE, severe PE, and HELLP. Samples were few in number in the present study. 

A significant positive association between PAPP-A—a known marker for PE, placental dysfunction, and FGR—and A/L ratio was found in PE pregnancies. This is interesting as the growth hormone—insulin-like growth factor (IGF) axis, where PAPP-A is a placenta-derived insulin-like growth factor binding protein (IGFBP)-2, -4, and -5 protease [66], plays a major role for both fetal and placental growth (possibly due to its importance for bioavailability of IGF) [67]. This specific finding needs to be confirmed through replication in other datasets.

A negative relationship between the A/L ratio and relative birth weight was only found in controls, suggesting that the homeostatic mechanisms involving adiponectin and leptin are perturbed in PE. This finding was independent of maternal BMI. One explanation could, however, be that 74% of PE pregnancies were induced, in contrast to 16% of controls. Thus, the timing of parturition is for the majority of PE cases determined by the obstetrician, and not by endocrine effector systems. The relationship between the A/L ratio and relative birth weight was also apparent when examining just AGA controls, indicating that it is not the extreme relative birth weights driving the overall association. Interestingly, the association between A/L ratio and relative birth weight in controls was driven by a positive association between leptin and relative birth weight. Adiponectin did not contribute to this association. Thus, relative birth weight does not seem to be dependent on IS, but rather on leptin-associated pathways [26]. 

Leptin contributed independently to the negative association between A/L ratio and relative birth weight in controls. Leptin has a permissive role in several reproductive functions [68], and in the current context, one might hypothesize that leptin reflects whether maternal energy stores are sufficient sustain normal fetal growth. We know that the relationship between leptin and maternal BMI is perturbed at high BMI levels [28], and increased IR originating from adipose tissue can be a contributing factor [69]. However, both leptin and adiponectin are involved in multiple physiological functions, e.g., immune- and metabolic regulation, so the precise definition of the mechanism behind the associations described here will have to be established through other types of studies [26].

PE is—in some cases—associated with reduced fetal size, and as expected, we found that women with PE had infants with significantly lower relative birth weight compared with controls. However, interestingly, there was a lack of association between birth weight and A/L ratio in PE pregnancies. These findings suggest that disturbances in metabolic homeostasis are already apparent long before the development of PE symptoms, by definition in week 20, and have negative consequences for fetal growth. Alternatively, the large abovementioned difference in induction rates between PE pregnancies and controls may over-ride the normal homeostatic mechanisms controlling the time of parturition. However, the period of relative IS and optimal nutrient access in the first trimester may be reduced in women who develop PE because they instead risk experiencing premature and/or increased IR. This possible shift between IS and IR in the first trimester may herald PE-associated impaired fetal growth. 

Even though this study included a broad range of clinical parameters, some important variables, such as the gender of offspring and gestational weight gain during pregnancy, were not included. Another limitation of this study was that only total adiponectin levels were measured. Three forms of adiponectin, low-, medium-, and high-molecular-weight (HMW) adiponectin, have been identified, where only HMW adiponectin has been shown to be associated with PE [70]. Finally, the A/L ratio was not compared with other known markers of insulin sensitivity/insulin resistance, such as HOMA-IR. Such a comparison might have helped substantiate the A/L ratio as a relevant marker of insulin sensitivity. 

## 5. Conclusions

A reduced A/L ratio is a first-trimester characteristic of pregnancies that later develop PE. However, neither severity nor time of onset was associated with A/L ratio. The A/L ratio, reflecting IS, is a better discriminator between PE and controls than either leptin or adiponectin. Relative birth weight was associated with A/L ratio; however, this was driven by leptin alone. Changes in birth weight do not seem to be dependent on insulin sensitivity, as is the case for PE, but rather on leptin-associated pathways. These findings may lead to novel paradigms towards the prevention of PE and low birth weight.

## Figures and Tables

**Figure 1 life-13-00130-f001:**
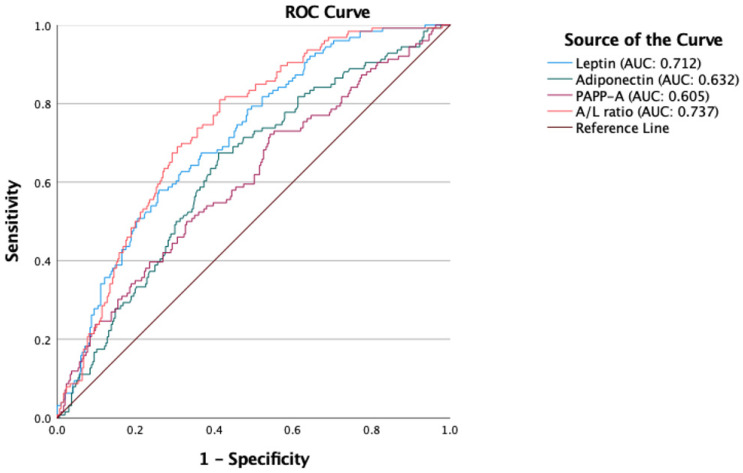
ROC curves of adiponectin, leptin, PAPP-A, and A/L ratio. A/L ratio was a better predictor of PE compared with adiponectin, leptin, and PAPP-A.

**Figure 2 life-13-00130-f002:**
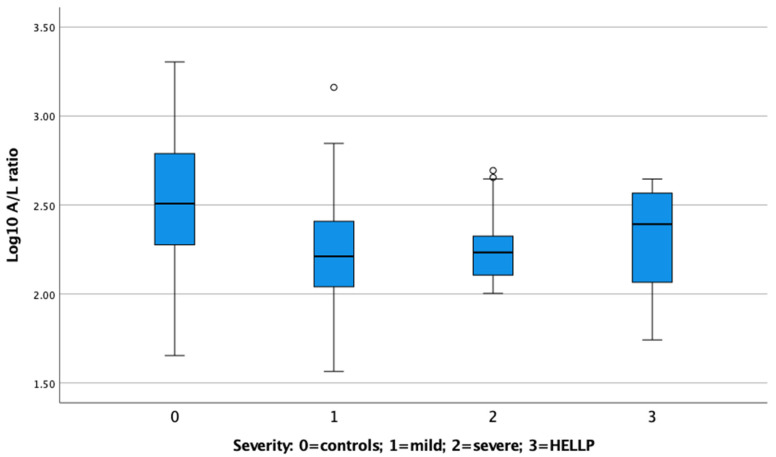
Box-plot of A/L ratio in controls, mild PE, severe PE, and HELLP. There was no discernable difference between A/L ratios in the different severity groups.

**Table 1 life-13-00130-t001:** Demographic and clinical characteristics of PE and control pregnancies.

Parameter	PE (*n* = 126)	Controls (*n* = 297)	*p*-Value
Maternal BMI (kg/m^2^), median (IQR)	24.0 (21.7–26.4)	21.7 (20.1–24.0)	<0.001
GA at birth (days), median (IQR)	275.5 (263.8–283.0)	284.0 (278.0–290.0)	<0.001
GA below 34 + 0 weeks, *n* (%)	10 (7.9)	0 (0.0)	<0.001
Crown rump length (mm), median (IQR)	67.00 (61.0–72.0)	67.0 (61.0–74.0)	0.54
Relative birth weight (%), mean (±SD)	91.39 (±14.98)	98.42 (±11.89)	<0.001
Weight classification, *n* (%)			<0.001
- SGA	18 (14.3)	6 (2.0)	
- AGA	104 (82.5)	284 (95.6)	
- LGA	4 (3.2)	7 (2.4)	
Nuchal translucency (mm), median (IQR)	1.6 (1.3–1.9)	1.5 (1.3–1.9)	0.51
GA at time of sampling (days), median (IQR)	90.5 (87.0–93.0)	90.0 (87.5–94.0)	0.63
A/L ratio, median (IQR)	0.17 (0.12–0.27)	0.32 (0.19–0.62)	<0.001
PAPP-A (mIU/L), median (IQR)	3064.0 (1941.3–4579.8)	3874.0 (2582.5–5979.5)	<0.001
HCGβ (IU/L), median (IQR)	42.4 (28.1–59.2)	41.8 (26.9–63.1)	0.94

Abbreviations: AGA: appropriate-for-gestational-age; A/L ratio: serum adiponectin/leptin ratio; BMI: body mass index; GA: gestational age; HCGβ: beta-chorionic gonadotropin; LGA: large-for-gestational-age; MoM: multiple of the median; PAPP-A: pregnancy-associated plasma protein A; SGA: small-for-gestational-age.

**Table 2 life-13-00130-t002:** Demographic, clinical, and paraclinical characteristics of mild PE, severe PE and HELLP pregnancies.

Parameter	Mild (*n* = 98)	Severe (*n* = 21)	HELLP (*n* = 7)	*p*-Value
Maternal BMI (kg/m^2^), median (IQR)	24.0 (21.6–27.2)	24.5 (22.4–26.1)	23.1 (21.9–26.7)	0.76
GA at birth (days), median (IQR)	278.0 (269.0–284.0)	264.0 (238.5–275.5)	254.0 (216.0–276.0)	<0.001
GA < 34 + 0, *n* (%)	2 (2.0)	5 (23.8)	3 (42.9)	<0.001
Crown rump length (mm), mean (±SD)	66.0 (±8.96)	68.9 (±8.67)	67.9 (±4.60)	0.37
Actual birth weight (g), median (IQR)	3295.0 (2899.0–3700.0)	2682.0 (1850.0–3255.0)	1629.0 (1475.0–3400.0)	0.002
Relative birth weight (%), mean (±SD)	92.8 (±13.91)	88.3 (±17.19)	80.5 (±18.82)	0.06
Weight classification, *n* (%)				0.02
- SGA	9 (9.2)	6 (28.6)	3 (42.9)	
- AGA	86 (87.8)	14 (66.7)	4 (57.1)	
- LGA	3 (3.1)	1 (4.8)	0 (0.0)	
Nuchal translucency (mm), mean (±SD)	1.6 (±0.49)	1.7 (±0.52)	1.3 (±0.36)	0.22
GA at time of sampling (days), median (IQR)	90.0 (86.0–93.0)	91.0 (88.5–94.0)	91.0 (90.0–92.0)	0.33
A/L ratio, median (IQR)	0.16 (0.11–0.26)	0.17 (0.13–0.24)	0.25 (0.07–0.37)	0.63
PAPP-A (mIU/L), median (IQR)	3260.5 (2341.0–5077.8)	2350.0 (1650.0–4427.5)	2283.0 (1753.0–2607.0)	0.08
HCGβ (IU/L), median (IQR)	43.1 (30.4–63.3)	31.2 (17.1–50.6)	43.2 (26.4–84.4)	0.17

Abbreviations: AGA: appropriate-for-gestational-age; A/L ratio: serum adiponectin/leptin ratio; BMI: body mass index; GA: gestational age; HCGβ: beta-chorionic gonadotropin; LGA: large-for-gestational-age; MoM: multiple of the median; PAPP-A: pregnancy-associated plasma protein A; SGA: small-for-gestational-age.

**Table 3 life-13-00130-t003:** Demographic, clinical, and paraclinical characteristics of early and late PE.

Parameter	Early PE (*n* = 10)	Late PE (*n* = 116)	*p*-Value
Maternal BMI (kg/m^2^), median (IQR)	25.6 (23.0–28.8)	24.0 (21.6–26.3)	0.20
GA at birth (days), median (IQR)	210.5 (201.0–229.8)	276.0 (268.0–283.8)	<0.001
PE severity, *n* (%)			<0.001
- Mild	2 (20.0)	96 (82.8)	
- Severe	5 (50.0)	16 (13.8)	
- HELLP	3 (30.0)	4 (3.4)	
Crown rump length (mm), median (IQR)	69.5 (62.3–72.3)	67.0 (60.3–72.0)	0.41
Relative birth weight (%), mean (±SD)	70.2 (±6.79)	93.2 (±14.07)	<0.001
Weight classification, *n* (%)			<0.001
- SGA	8 (80.0)	10 (8.6)	
- AGA	2 (20.0)	102 (87.9)	
- LGA	0 (0.0)	4 (3.4)	
Nuchal translucency (mm), median (IQR)	1.7 (1.4–1.8)	1.5 (1.3–1.9)	0.98
GA at time of sampling (days), median (IQR)	90.5 (87.5–93.0)	90.5 (87.0–93.0)	0.85
A/L ratio, median (IQR)	0.13 (0.11–0.25)	0.17 (0.12–0.27)	0.45
PAPP-A (mlU/L), median (IQR)	1761.0 (1470.0–2194.3)	3239.5 (2291.5–5019.8)	0.003
HCGβ (IU/L), median (IQR)	42.4 (28.1–59.2)	42.4 (29.6–61.9)	0.21

Abbreviations: AGA: appropriate-for-gestational-age; A/L ratio: serum adiponectin/leptin ratio; BMI: body mass index; GA: gestational age; HCGβ: beta-chorionic gonadotropin; LGA: large-for-gestational-age; MoM: multiple of the median; PAPP-A: pregnancy-associated plasma protein A; PE: pre-eclampsia; SGA: small-for-gestational-age.

## Data Availability

The datasets generated and/or analyzed during the current study are not publicly available due to restrictions in handling personal data, and particularly because participants were recruited at a time when it was not standard practice to make samples/data publicly available but meta-data are available from the corresponding author on reasonable request.

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
