# Peer review of "First-Trimester Maternal Serum Adiponectin/Leptin Ratio in Pre-Eclampsia and Fetal Growth"

_life, 2023, doi:10.3390/life13010130_

Round 1

Reviewer 1 Report

Because on relation wan observed between the A/L ratio and fetal growth in PE, I think the title of the manuscript should be changed.

It would have been better if another marker of insulin resistance such as HOMA-IR was measured and compared with the A/L ratio.

Author Response

Reviewer 1

We thank for excellent comments and have made the following changes accordingly:

  1. Because on relation wan observed between the A/L ratio and fetal growth in PE, I think the title of the manuscript should be changed.

Response: This is a very relevant relevant suggestion. In this paper, we conclude that an impairment of the serum adiponectin/leptin ratio (A/L ratio) in first-trimester is characteristic of PE, while aberrant fetal growth in PE is not dependent on insulin sensitivity but rather on leptin associated pathways. Thus, the original title is misleading. We have now chosen the more neutral title: “First trimester adiponectin/leptin ratio in pre-eclampsia and fetal growth”.

  1. It would have been better if another marker of insulin resistance such as HOMA-IR was measured and compared with the A/L ratio.

Response: We agree that it would have improved the paper if the use of the A/L ratio as a surrogate marker of insulin sensitivity could have been supported by other markers of insulin resistance, e.g. HOMA-IR. Unfortunately, the necessary data on insulin and fasting glucose were not available. Instead we have added the two sentences “Finally, the A/L ratio was not compared to other known markers of insulin sensitivity/insulin resistance, such as HOMA-IR. Such a comparison might have helped substantiate the A/L ratio as a marker of insulin sensitivity” to the Discussion (lines 378-380) acknowledging this weakness. The lack of HOMA-IR data also precludes a more detailed discussion of the relation between the A/L ratio and insulin resistance/sensitivity – however a paper by Inoue et al 2005 (ref 51 in our manuscript) suggests that A/L ratio might be a better marker for insulin sensitivity, as it is not as sensitive to high fasting glucose. So, as suggested by the reviewer, a comparison of various markers of insulin sensitivity would be of great interest – however it is, from lack of data, out of the scope of the present manuscript.

 Yours sincerely,

Michael Christiansen

Reviewer 2 Report

The study is well planned and nicely presented. However, the AUC signifies only a fair association of A/L ratio with PE. The test doesn't qualify to be used as a predictor of PE. 

In my opinion, placenta and fetal growth restriction can be removed from keywords as these words do not appear in the abstract. Instead, A/L ratio can be included. 

Author Response

Reviewer 2

We thank for constructive comments and have changed the manuscript accordingly as detailed below: 

  1. The study is well planned and nicely presented. However, the AUC signifies only a fair association of A/L ratio with PE. The test doesn’t qualify to be used as a predictor of PE.

Response: We fully agree with the reviewer that the AUC does not signify a strong association of the A/L ratio and PE and has no utility as a first-trimester stand-alone predictor of PE. We have stated in the discussion (lines 316-317) that the AUC of A/L ratio is so small that it precludes a use of A/L ratio as a first-trimester single marker of later PE. In lines 319 -321 we do discuss the possible utility of the A/L ratio in PE screening algorithms, but only in combination with other clinical or paraclinical predictors of PE. However, we realise that we have stated in line 46 of the abstract, line 235 of the results, and line 294 and 316 of the discussion that the A/L ratio was a better predictor of PE than leptin and adiponectin alone, why we have added the half-sentence: “.., albeit not clinically relevant as a single marker” to the abstract line 46.   

  1. In my opinion, placenta and fetal growth restriction can be removed from keywords as these words do not appear in the abstract. Instead, a/l ratio can be included.

Response: We completely agree and have included A/L ratio in the list of keywords and have removed placenta and fetal growth restriction.

Yours sincerely

Michael Christiansen

Reviewer 3 Report

The Introduction section along with the rationale of the paper is sound and well written leading to well defined objectives. The paper offers interesting results, the use of English is very good and manuscript is professionally presented. However, the author should mention here how the present work adds to the literature and highlight the strength of the study.

 1.     Introduction

Please comment on the changes in adiponectin and leptin levels as pregnancy progresses.

 As the associations reported by the author are already described in other papers, it is important to highlight what this data adds to the literature. Please refer to the recent article by Rao et al., 2021(https://www.ncbi.nlm.nih.gov/pmc/articles/PMC7960875/ )

 2.     Methods

Does control group included preterm infants and low birthweight infants. Did they separately look at whether gestational age and birthweight related to BP before adjusting for these factors?

 How did the authors verify mothers classed as ‘control’ did not have hypertension in other pregnancies.

 3.     Discussion

Comment on the mechanistic aspect of adipocytokines on insulin sensitivity in pregnancy in discussion.

 4.     Conclusion

It is recommended to highlight implications of the study in the conclusion.

Author Response

Reviewer 3

 We thank for insightful and cnstructive comments and have changed the manuscript accordingly, as detailed below:

The introduction section along with the rationale of the paper is sound and well written leading to well defined objectives. The paper offers interesting results, the use of English is very good and manuscript is professionally presented. However, the author should mention here how the present work adds to the literature and highlight the strength of the study.

Response: We thank for the fine overall assessment of our manuscript and have made the following changes to accommodate the reviewer request of a stronger emphasis of what is special and new in our manuscript compared to the literature: 1. The title has been changed to “First trimester maternal serum adiponectin/leptin ratio in pre-eclampsia and fetal growth”; 2. The word “First-trimester” is added to the Introduction (line 98), and 3. The sentence “This is compatible with the A/L ratio being a diagnostic marker of PE in late pregnancy (51,52)” has been added to paragraph 1 of the Discussion (line 299-300). Hereby we find that we have emphasized further that we are studying the levels of A/L in first trimester and not in later parts of pregnancy. And this approach is novel.

  1. Introduction

Please comment on the changes in adiponectin and leptin levels as pregnancy progresses.

As the associations reported by the author are already described in other papers, it is important to highlight what this data adds to the literature. Please refer to the recent article by Rao et al., 2021 (hhtp://www.ncbi.nlm.nih.giv/pmc/articles/PMC7960875)

Response: We agree that the normal levels of adiponectin and leptin during pregnancy are important to inform about and in the Introduction we have added the sentences “..that exhibit decreased maternal serum levels in pregnancy (24),..”(line 112) and “…where the maternal serum levels increase through pregnancy to normalize after birth (25)”(line 113), describing the pregnancy levels of adiponectin and leptin, respectively. The references 24 and 25 are added. The paper by Rao et al has been added to the Discussion (line 300)as reference 52, as it is particularly relevant for the finding that A/L is an informative marker in late pregnancy.

  1. Methods

Does control group included preterm infants and low birthweight infants. Did they separately look at whether gestational age and birth weight related to BP (blood pressure) before adjusting for these factors?

Response: We recognize the importance of a careful description of patients as well as controls in the type of study performed here. In consequence, we present demographic/clinical data in Table 1. Here we include data on GA below 34+0 weeks and weight classification using intrauterine growth curves developed by Marsal et al. where infants were classified as SGA, AGA and LGA.

No infants in the control group were born <GA 37. Six of 297 controls were born SGA. We did not separately analyse the significance of the BP – in particular because the quality of determination of BP was not assessed. We only used the clinical diagnosis PE with proteinuria. We agree that a new study should very carefully register BP under standard conditions. However, this was not done in our study.

  1. How did the authors verify mothers classed as ‘control’ did not have hypertension in other pregnancies.

Response: We ascertained that the controls did not suffer from hypertension, chronically or in the present pregnancy. However, it was not possible for us to get information about hypertension during previous pregnancies. Of the mothers in the control group 44.4% had been pregnant before the current pregnancy, so the proportion of controls with previous hypertension – and not actual, that was excluded – must be expected to be very low, max 1-2% of controls. However, several risk factors were not accounted for, except through random selection outside the selection criteria, but this was the conditions inherent in our study. The limited number of PE pregnancies will also make it difficult to compensate for many factors. In consequence, the clinical applicability of our findings must await further studies, preferably prospective studies.

  1. Discussion

Comment on the mechanistic aspect of adipocytokines on insulin sensitivity in pregnancy in discussion.

Response: We acknowledge that the physiological roles of adiponectin and leptin in pregnancy are very complex with many effector systems and functions involved. So, our findings do not really support statements beyond the association with insulin sensitivity and birth weight. To emphasize this we have added the sentence “However, both leptin and adiponectin are involved in multiple physiological functions, e.g. immune- and metabolic regulation, so the precise definition of the mechanisms behind the associations described here will have to be established through other types of studies” to the Discussion (line 362 – 365).

Yours sincerely

Michael Christiansen